# SARS-CoV-2 Antigen Testing Intervals: Twice or Thrice a Week?

**DOI:** 10.3390/diagnostics12051039

**Published:** 2022-04-21

**Authors:** Chin Shern Lau, Tar-Choon Aw

**Affiliations:** 1Department of Laboratory Medicine, Changi General Hospital, Singapore 529889, Singapore; mike.lau.cs@gmail.com; 2Department of Medicine, National University of Singapore, Singapore 119077, Singapore; 3Academic Pathology Program, Duke-NUS Medical School, Singapore 169857, Singapore

**Keywords:** SARS-CoV-2, antigen, frequency

## Abstract

Antigen testing for SARS-CoV-2 has become an increasingly prominent screening tool in the ongoing COVID-19 pandemic and can be performed multiple times a week. However, the optimal weekly frequency of antigen testing is unclear; the Centers for Disease Control and Prevention recommends 1–3 times a week, while some experts support testing 2–3 times a week. In our own laboratory, all staff (n = 161) underwent twice- and thrice-weekly antigen tests during different periods from August 2021 to the present as part of routine COVID-19 surveillance of healthcare workers. No cases of COVID-19 were detected with either regimen. While more frequent SARS-CoV-2 antigen testing may allow antigen testing to be an important surrogate for RT-PCR testing, performing SARS-CoV-2 antigen tests twice or thrice a week shows no inferiority to each other in screening for COVID-19.

## 1. Introduction

Antigen testing for SARS-CoV-2 has become an increasingly prominent screening tool in the ongoing COVID-19 pandemic. Indeed, when used to screen patients in an emergency department in a high prevalence setting, rapid antigen testing (LumiraDx SARS-CoV-2 Antigen test) had a sensitivity of 68.7% when compared to RT-PCR testing; sensitivities increased to 81% when confined to symptomatic patients only [1]. Another automated antigen detection system showed a 95.4% concordance with RT-PCR in samples with cycle threshold (Ct) count numbers of >100 [2]. We recently reported on the performance of the Roche Elecsys SARS-CoV-2 antigen high-throughput assay [3], which had good sensitivity (94.7%) in COVID-19 subjects with Ct counts of ≤30. Another study [4] using an immunofluorescence immunoassay analyzer (Sofia, Quidel) demonstrated improved sensitivities (90%) for symptomatic patients ≤ 5 days versus 82% >5 days post symptom onset. This correlation also extends to lateral flow immunoassay (LFIA) antigen tests. Indeed, one study compared 14 different rapid antigen tests, performing 400 evaluations of RT-PCR positive samples and 50 evaluations of RT-PCR negative samples [5]. All rapid tests displayed a reverse correlation of the colorimetric intensity of the band with RT-PCR Ct count values, despite high variability in performance between assays, with total agreement between all 14 assays at Ct thresholds of <27 (89.9% agreement when Ct < 30). The combined overall sensitivity of all the 14 LFIAs was 74.3% and improved to 88.2% when confined to Ct counts of ≤30. Thus, SARS-CoV-2 antigen testing, quantitative or qualitative, can be used to detect new COVID-19 infections, especially in symptomatic patients with recent disease onset and higher viral loads.

Although RT-PCR testing remains the gold standard for diagnosing SARS-CoV-2 infection, it is expensive to perform, and requires a significantly longer time to report results. The advantage of antigen testing is that it is cheaper and easier to perform than RT-PCR testing and can provide results in as little as 15–20 min. Thus, antigen tests have gained widespread use in screening programs across the world. However, some screening programs only report findings based on the use of a single antigen test only. For example, one recent report [6] performed testing on 991 subjects using the Standard Q COVID-19 LFIA (SD Biosensor) and the LIAISON SARS-CoV-2 antigen chemiluminescence immunoassay, but only tested each sample once on each assay. Predictably, the sensitivity of both assays was less than RT-PCR testing, with the LIAISON only having a sensitivity of 43.3% and the Standard Q only 30.6% compared to RT-PCR. The most optimal way to utilize SARS-CoV-2 antigen tests is to take advantage of their ability to be performed multiple times a week, given their ease of use. This would allow even less sensitive rapid antigen tests to outperform more sensitive RT-PCR tests in detecting COVID-19 earlier in the infectious window [7]. Even in mathematical models, tests with low sensitivity, administered daily, can outperform high sensitivity tests, with the total number of cases over a 6-month period reduced from 19.4% (RT-PCR returned in 24 h) to less than 1.22% with a rapid test performed daily, and 3.8%/5.4% when performed every other day/every three days [8]. However, different screening programs have adopted variable time intervals for antigen testing, ranging from weekly to daily regimens. Indeed, the Centers for Disease Control and Prevention has now recognized this variability and recommends that for symptomatic patients or asymptomatic persons with contact with suspected cases, serial antigen testing should be performed every 3–7 days for 10 days after an initially negative result [9]. Furthermore, not all positive results on antigen screening tests may be true positives, especially in areas of low prevalence. In one reported screening program, performed in a university for over 2 months [10], weekly antigen testing detected 133 positive antigen tests out of 10,360 samples from asymptomatic individuals; however, follow-up RT-PCR testing only showed 35 true positives. Additional measures are required to improve the false-positive rate. We report our experience with screening regimens using two different testing intervals of 2 or 3 times per week.

## 2. Methods

Both the SD Biosensor Standard Q [11] and Abbott Panbio COVID-19 Ag Rapid Tests [12] are qualitative LFIAs. The Abbott Panbio has a reported sensitivity of 98.1% and specificity of 99.8%, with sensitivity increasing to 99.0% in samples with Ct values of ≤33. The Standard Q LFIA has a manufacturer-reported sensitivity of 82.7%, and specificity of 99.1%. In samples with Ct counts ≤30, professionally collected samples had a sensitivity of 89.6%. Both are membrane-based immunochromatographic assays which detect the SARS-CoV-2 nucleocapsid protein. After sampling from both nostrils, swabs are placed in dedicated collection tubes with a sampling buffer. Results are available within 15–30 min, as indicated by a positive test line with a positive control line. For RT-PCR testing, our hospital molecular laboratory employs a duplex real-time RT-PCR that targets the N and E genes using a Qiagen EZ1 extraction system and using the Cobas SARS-CoV-2 qualitative assay on the Cobas 6800 System.

## 3. Results

Our laboratory has 161 staff members. As part of routine rostered testing, all staff has been tested with twice-weekly rapid antigen testing. In August 2021, one of our staff members (immunized with 2 doses of Pfizer BNT162b2 mRNA vaccine) developed COVID-19 symptoms and tested positive for SARS-CoV-2 on RT-PCR. Subsequently, all staff members underwent immediate RT-PCR testing every three days (Roche SARS-CoV-2 qualitative assay on the Cobas 6800), and rapid antigen testing (Standard Q, SD Biosensor and Abbott Panbio COVID-19 Ag Rapid Nasal Test Device) every day for the next two weeks as previously reported [13]. As no new cases were detected, RT-PCR testing was discontinued, and all staff reverted to rapid antigen testing twice a week until the end of November 2021. With an increasing number of COVID-19 cases in our hospital staff, regular screening with rapid antigen tests was increased to three times a week in the hope of improving detection rates. This regimen was applied until the 10th of December 2021. We since reverted to twice-weekly testing until present (the timeline is displayed in Figure 1). Booster shots of the Pfizer mRNA were administered to staff from the end of September 2021 onwards. Over the entire period (August 2021 to the present), none of our staff tested positive for SARS-CoV-2 with either twice- or thrice-weekly antigen testing.

## 4. Discussion

In COVID-19 screening programs that use antigen testing, the optimal time interval between tests is an open question. Studies on viral kinetics seem to offer some insight into this conundrum. In a recent study [14] where volunteers were inoculated with a wild-type virus intranasally, viral loads rose steeply and peaked at 5 days post-inoculation, with symptoms beginning 2–4 days and viral shedding around 2 days post-inoculation. LFIA results were strongly associated with the viral loads, with all infected individuals having positive LFIAs ≥ 2 days post-inoculation, with a median time to detection of 4 days when using daily LFIA tests. In their modeling data, twice-weekly LFIA testing could diagnose infection before 70–80% of the viable virus was generated, further supporting the validity of performing LFIAs roughly every 3 days.

This finding has been confirmed in another study [15] where RT-PCR positive cases (n = 43) were followed for up to 14 days in quarantine with daily salivary and nasal swabs for RT-PCR, Sofia SARS antigen fluorescence immunoassay, and viral cultures. For all 3 tests, sensitivity remained >98% when tested at least every third day. Daily antigen testing at any stage of infection had a sensitivity of 100% when performed every other day or every third day, but declined to 79.7% when only used once weekly. This demonstrates the non-inferiority of frequent antigen tests to RT-PCR tests in COVID-19 screening. Indeed, one study [16] estimated disease transmission rates based on a fully mixed model of 20,000 individuals with zero initial infections and a constant 1/N per-person probability of becoming infected from an external source. They predicted that population screening with self-testing every three days can achieve an approximately 40% reduction in disease reproduction numbers, compared to weekly testing. Another modeling study [17] used a simple compartmental epidemic model, based on a hypothetical cohort of 5000 unvaccinated college students followed over a period of 80 days using a test that was 70% sensitive and 98% specific. They found that twice- and thrice-weekly testing led to 379 and 243 cumulative infections, respectively, compared to 1840 cumulative infections with weekly screening (assuming a reproduction number of 2.5, with 10 exogenous infections each week). Of note, the incremental cost-effectiveness ratio of twice-weekly testing proved superior to thrice-weekly testing, at $600 per infection averted compared to $5700 per infection averted. Another mathematical modeling study [18] compared the performance of a low sensitivity test (60% sensitivity) performed twice weekly in low (constant rate of introduced infections) and high (high-growth external community prevalence) community prevalence settings, in a relative population of 1500 individuals (assumed reproduction number of 2.5, average period of infectiousness 4.5 days). This resulted in a 58.2% and 48.9% reduction in cumulative infections. Together with our findings, these studies lend support to the notion that twice or thrice weekly strategies are non-inferior to each other; in fact, twice-weekly antigen testing may be sufficient. It is also important to note that the performance of these LFIAs is dependent on the user’s proficiency in performing the test on these home-based test kits. Thus, the users must familiarize themselves with the instructions for use of the test kits. The staff in our laboratory have been performing self-administered LFIAs from May 2021 to March 2022, and with increased frequency of testing (of at least twice a week), positive cases would be more likely to be detected. Furthermore, both the Panbio and SD Biosensor LFIAs have been evaluated in the latest Cochrane review [19].

The most appropriate use for frequent rapid antigen testing would be in the detection of early COVID-19 infection when patients have the highest viral loads (lower Ct counts) or are just becoming symptomatic. Indeed, twice-weekly antigen testing has excellent sensitivity (96.2%) when compared with RT-PCR testing in early disease (days 0 to 3 of symptom onset) [20]. A meta-analysis [21] also demonstrated that rapid antigen tests had superior sensitivities in patients with RT-PCR Ct counts ≤ 25 (sensitivity of 96%, compared to 69% if >25) and in symptomatic individuals (sensitivity 82%, compared to 68% in asymptomatic subjects). In another real-world study [22] of 10 rapid antigen test kits, the overall sensitivity was greater in samples with Ct counts ≤ 25 than in samples with counts of ≤30 (88.1% vs. 76.9%), with the Vazyme SARS-CoV-2 Antigen Detection Kit having the greatest sensitivity (72.0%) and the Viva-Diag SARS-CoV-2 Ag Rapid Test having the lowest sensitivity (43.0%). In a similar vein, antigen testing was 90% sensitive in subjects with positive viral cultures [15], with sensitivity peaking during the days in which infectious virus shedding was detectable.

The most optimal method of screening for COVID-19 would be to use both antigen and RT-PCR testing in an orthogonal fashion, especially in settings of low disease prevalence to minimize false-positive antigen results. All “positive” antigen tests would be retested with RT-PCR and only those positive on both tests would considered true positives. Low disease prevalence would greatly reduce the positive predictive value of any individual test, leading to an increase in false-positive results. Orthogonal testing can greatly reduce the false positive rate [23]. This is supported by reports in the real-world setting; RT-PCR testing with twice-weekly LFIAs over a period of 3 weeks ruled out 11 false-positive antigen tests (0.007%) in 156,000 subjects (20 true-positive cases, 0.013%) [24]. In another report [25], RT-PCR testing of positive cases from twice weekly LFIA testing detected 462 false-positive (0.05%) results in 903,408 rapid antigen tests, out of 1322 initially positive results (0.15%). Such examples inform the ability of orthogonal testing to reduce false-positive cases effectively and avoid unnecessary patient treatments and admissions.

The strength of our study is that we compared twice- vs. thrice-weekly COVID-19 screening using LFIAs in a disease-free population, which supplements studies that compared twice vs. thrice weekly testing in infectious cases [15], or reports that compare daily vs. twice-weekly antigen testing [13]. Our laboratory staff commute from their homes to work each day. In the community in Singapore, we faced rising COVID-19 from the ascent of delta cases. Daily confirmed COVID-19 cases per million people rose from 100 in mid-September to peak at 650 by the end of October before declining to 50 towards the end of December 2021 (data from the John Hopkins University CSSE COVID-19 database at ‘Our World in Data’). In fact, the first booster doses began to be administered in mid-September. During the period of our study, no cases were detected amongst our staff in the face of rising cases in the community. This fact of testing frequency has been clearly described in the literature [15] when daily PCR is compared to daily antigen testing. When one of our staff members contracted COVID-19 in August 2021, all lab staff were required by our hospital infection control team to undergo daily antigen testing and PCR on days 1, 4, 7, and 10 [13]. Our hospital authorities again mandated antigen testing 3 times a week from the end of November for 2 weeks before reverting to twice-weekly antigen testing. While antigen testing 2 or 3 times a week may be equivalent in the face of significant cases in the community the outcomes are probably a reflection of good personal hygiene, effective public health measures, and vaccine boosters.

One limitation of our study is that two different rapid antigen testing kits were used by our staff over the time period, due to logistical supply constraints at our institution. Despite the varying positive predictive values of both tests differing at various levels of disease prevalence (see Table 1), we did not detect any positive cases on both tests over the entire study period, as both have high test specificities of over 99%. Furthermore, Table 1 also demonstrates that the negative predictive value of both tests is similar even with variable disease prevalence, although a small change in specificity would result in quite a drastic change in positive predictive values.

## 5. Conclusions

More frequent SARS-CoV-2 antigen testing can allow antigen testing to be an important surrogate for RT-PCR testing. However, performing antigen testing twice or thrice a week did not detect any new cases; thus, two tests per week may suffice.

## Figures and Tables

**Figure 1 diagnostics-12-01039-f001:**
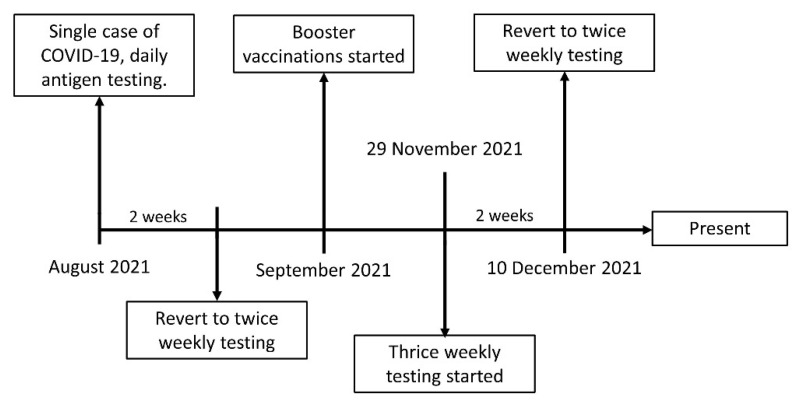
Timeline of SARS-CoV-2 rapid antigen testing from August 2021 to present.

**Table 1 diagnostics-12-01039-t001:** The effect of disease prevalence on the positive and negative predictive values of the Panbio and Standard Q rapid antigen tests.

Prevalence	Panbio (Sensitivity 98.1%, Specificity 99.8%)		Standard Q (Sensitivity 82.7%, Specificity of 99.1%)	
	Positive Predictive Value	Negative Predictive Value	Positive Predictive Value	Negative Predictive Value
0.1%	32.90%	100%	8.40%	100%
0.5%	71.10%	100%	31.60%	99.9%
1.0%	83.20%	100%	48.10%	99.8%

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
