# Peer review of "SARS-CoV-2 Antigen Testing Intervals: Twice or Thrice a Week?"

_diagnostics, 2022, doi:10.3390/diagnostics12051039_

Round 1
Reviewer 1 Report
The authors described in this paper SARS COV-2 antigen testing. The underlined the material and methods used to test the COVID disease. The article is well organized and structured. In my opinion, it is suitable to Diagnostics publication. Minor revisions are needed.
Minor Points
Q1. The authors must described accurately the Material and Methods used in the manuscript. They focus on the methods and the primers used.
Q2. The authors must improved the discussion.
Author Response
Reviewer 1
Comment 1: The authors must describe accurately the Material and Methods used in the manuscript. They focus on the methods and the primers used.
Reply: We have added a statement on the RT-PCR methods and primers used in the Methods section.
Comment 2: The authors must improve the discussion.
Reply: We have expanded the discussion (yellow highlight).
Reviewer 2 Report
Thank you for the communication. It would be interesting to highlight any additional results or insight the current article brings beyond what has already been published (your reference no.11). How to sustain the article conclusions given a situation of zero COVID incidence during the study period?
Author Response
Reviewer 2
Comment 1: It would be interesting to highlight any additional results or insight the current article brings beyond what has already been published.
Reply: Reference [11] now [13] was conducted in August-September 2021, and compared daily vs twice weekly testing. Thus we now provide additional information by comparing twice vs thrice weekly COVID-19 screening using LFIAs from October 2021 to March 2022. We have now reflected this in the discussion.
Comment 2: How to sustain the article’s conclusions given a situation of zero COVID incidence during the study period?
Reply: We sought to describe the screening program performance in a disease-free cohort. Even though community COVID-19 cases continued to increase, our staff still did not contract COVID-19, thanks to strict use of PPE and safe distancing measures. As such, we can conclude that the performance of twice vs thrice weekly screening was similar.
Reviewer 3 Report
Authors must add visualization results to paper
The collected data must be uploaded to check it
Author Response
Reviewer 3
Comment 1: Authors must add visualization results to paper.
Reply: We have now separated the methods and results into two different sections, to improve the visualization of the results.
Comment 2: The collected data must be uploaded to check it.
Reply: Rapid antigen tests are qualitative and reported as positive or negative. All data has already been described in the article text.
Round 2
Reviewer 2 Report
The comparison of 2 vs. 3 times/wk testing in a condition of NO transmission seems inconclusive from an epidemiological and clinical study design perspectives. The agreement of the negative predictive value of 2 vs. 3 tests is 100% (no false positive), but nothing can be said about the positive predictive value in a situation when there is transmission.
Author Response
The staff do not live onsite in the period under study. In fact they commute from their homes to the lab each day. In the community in Singapore we faced rising cases from the ascent of delta cases. Daily confirmed Covid19 cases per million people rose from 100 in mid-September to peak at 650 by end October before declining to 50 towards end December 2021 (see accompanying chart from the John Hopkins University CSSE Covid19 database at 'Our World in Data'). In fact the first booster doses began to be administered in mid September. The fact that no cases were detected in the face of rising cases in the community is clear. Whether antigen testing was employed 2 or 3 times a week is probably a reflection of good personal hygiene, effective public health measures and vaccine boosters. This fact of testing frequency has been clearly described in reference 15 (Smith et al) when compared to daily PCR or daily antigen testing. When one of our staff members contracted CoVid 19 in August 2021 all lab staff were required by our hospital infection control team to undergo daily antigen testing and PCR on days 1,4,7, and10 (Lau et al. reference 13). Our hospital authorities again mandated antigen testing 3 times a week from end November for 2 weeks before reverting to twice weekly antigen testing. We wish to report our experience that testing 2 or 3 times a week are probably equivalent in the face of quite significant cases in the community and has already been tested in the Smith paper (reference 15).

Round 3
Reviewer 2 Report
The notes to report 2 provided by the authors helps to understand the rationale and conclusions of the study and should be included in the article.
However, the wording and conclusion statement that testing 2 or 3 times a week “are probably equivalent” needs to be modified, saying that even testing 2 or 3 times a week the laboratory did not diagnose a new case.
Author Response
Thank you.
Agree.
All suggestions incorporated.
Author Response:
12 April 2022
To the editor,
We have noted the reviewer’s comments and incorporated the suggestions accordingly:
Reviewer 2
The notes to report 2 provided by the authors helps to understand the rationale and conclusions of the study and should be included in the article.
Done.
However, the wording and conclusion statement that testing 2 or 3 times a week “are probably equivalent” needs to be modified, saying that even testing 2 or 3 times a week the laboratory did not diagnose a new case.
Done.
We thank the reviewers for their constructive suggestions that have strengthened the manuscript.
Regards,
Prof Aw Tar Choon
Dr Michael Lau